# LLaVA-Interleave:
# Tackling Multi-image, Video, and 3D in Large Multimodal Models

**Feng Li**[1,2*], **Renrui Zhang**[1,3*], **Hao Zhang**[1,2*], **Yuanhan Zhang**[1,4],
**Bo Li**[1,4], **Wei Li**[1], **Zejun Ma**[1], **Chunyuan Li**[1]

[1] ByteDance  [2] HKUST  [3] CUHK  [4] NTU          [*] Core contributor

https://llava-vl.github.io/blog/2024-06-16-llava-next-interleave/

Figure 1: Performance comparison in three interleaved scenarios, including multi-image, multi-frame (video), and multi-view (3D). Our LLaVA-Interleave model achieves SoTA performance across a variety of evaluation benchmarks.

## Abstract

Visual instruction tuning has made considerable strides in enhancing the capabilities of Large Multimodal Models (LMMs). However, existing open LMMs largely focus on single-image tasks, their applications to multi-image scenarios remains less explored. Additionally, prior LMM research separately tackles different scenarios, leaving it impossible to generalize cross scenarios with new emerging capabilities. To this end, we introduce **LLaVA-Interleave**, which simultaneously tackles **M**ulti-image, **M**ulti-frame (video), **M**ulti-view (3D), and **M**ulti-patch (single-image) scenarios in LMMs. To enable these capabilities, we regard the interleaved data format as a general template and compile the **M4-Instruct** dataset with 1,177.6k samples, spanning 4 primary domains with 14 tasks and 41 datasets. We also curate the **LLaVA-Interleave Bench** to comprehensively evaluate the multi-image performance of LMMs. Through extensive experiments, LLaVA-Interleave achieves leading results in multi-image, video, and 3D benchmarks, while maintaining the performance of single-image tasks. Besides, our model also exhibits several emerging capabilities, e.g., transferring tasks across different settings and modalities. Code is available at https://github.com/LLaVA-VL/LLaVA-NeXT.

# 1 Introduction

Recent advancements in Large Multimodal Models (LMMs) (37; 26; 43; 12; 64; 11; 66) have showcased impressive capabilities in diverse multimodal contexts, advancing the pursuit of artificial general intelligence. With extensive vision-language data (46; 47), they empower Large Language

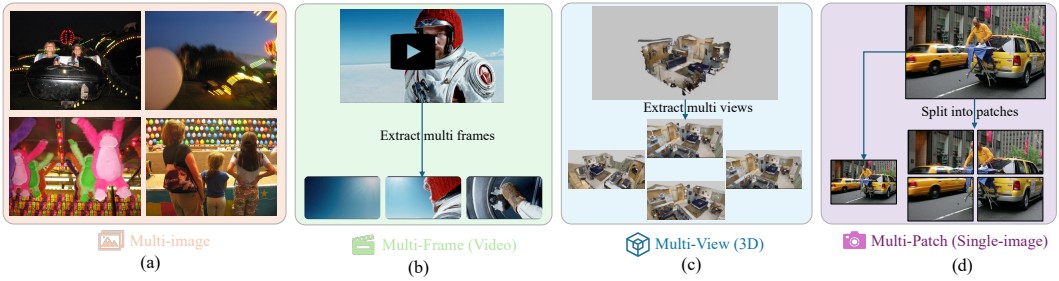

Figure 2: Tasks in our M4-Instruct. (a) showcases an example of interleaved multi-image scenarios (visual story telling). (b), (c), and (d) indicate that video, 3D and single-image data can also be organized as the interleaved data format for unified processing.

Models (LLMs) (52; 53; 8; 5) with visual modality by aligning vision encoders (44; 9; 45). This integration has propelled forward the field of AI, enabling complex image and language understanding tasks to be performed with unprecedented accuracy.

However, most open-source LMMs (24; 11; 34; 36) have primarily focused on pushing the performance limit of the single-image scenario, the more complex multi-image scenarios remain largely less explored. This oversight is significant given that many real-world applications demand multi-image capabilities, such as comprehensive multi-image analyses. Traditionally, researchers have approached these challenges by training separate task-specific models for each application scenario, e.g., multi-image (19; 27; 1), video (7; 29; 67), and 3D (14; 58; 15). This is both labor-intensive and time-consuming, resulting in fragmented methodologies that are inefficient and often unscalable. Considering the diverse range of computer vision settings and data formats, there is a pressing need to develop a general framework for LMMs that can operate effectively across these varied contexts.

In this paper, we observe that the image-text interleaved format can naturally serve as a general data template to unify different scenarios, e.g., single-image or multi-image as special cases, video as multi-frames, and 3D as multi-views, as illustrated in Figure 2. Therefore, we present *LLaVA-Interleave*, an all-around LMM that extends the model capabilities to various real-world settings such as *M*ulti-image, *M*ulti-frame (videos), *M*ulti-view (3D) while maintains the performance of the *M*ulti-patch (single-image) performance. We denote the four settings as *M4*.

The core innovation of our approach lies in the perspective to leverage an image-text interleaved format as a universal data template capable of accommodating different scenarios, and construct the related visual instruction-following data. This perspective not only simplifies the training process across various domains, but also allow the model to emerge new capabilities due to cross-domain task composition.

Our contributions are summarized as below:

- ***Interleave data format unifies different tasks.*** We convert multi-image, video, 3D, and single-image data all into an interleaved training format, which unifies different tasks in a single LMM.

- ***New dataset and benchmark.*** We compile a high-quality training dataset, **M4-Instruct**, with 1177.6 samples to empower LMMs with the M4 capabilities, which spans 4 primary domains (multi-image, video, 3D, and single-image) with 14 tasks and 41 datasets. We also curate LLaVA-Interleave Bench, a diverse set of benchmarks to evaluate the multi-image performance, including 7 newly collected and 13 existing in/out-domain benchmarks.

- ***SoTA performance.*** With a single model, LLaVA-Interleave can achieve leading results across different multi-image tasks compared to the previous SoTA, while maintaining the single-image performance, as exemplified in Figure 1.

- ***Emerging capabilities with cross-task transfer.*** By jointly training on a diverse set of tasks, our model showcases emerging capabilities to transfer tasks across different settings and modalities. e.g., from spotting differences between images to videos.

## 2 RELATED WORK

**Interleaved Image-text Training Data.** As a more general format, interleaved image-text data can enable LMMs with two distinctive capabilities: multimodal in-context learning (ICL) capability and instruction-following capability in real-world multi-image application scenarios. *The former in-context scenarios* interleave several image-text examples within the prompt as task demonstrations, adapting LMMs to new tasks in the inference stage in a few-shot manner. Flamingo (1) is first model to demonstrate this capability, and thus is considered as GPT-3 moment for multimodal community. Typically, the multimodal ICL ability is emerged after pre-training on web-scale raw interleaved image-text sequences. In the open-source community, MMC4 (68) introduces a public 101.2M interleaved dataset spanning everyday topics, OBELICS (22) also presents a filtered dataset comprising 141M interleaved web pages. Kosmos-1 (18) curates a 71M multimodal corpora, including arbitrarily interleaved documents. To explicitly enable the ICL capability, MIMIC-IT (25) proposes an automatic pipeline to create 2.8M multimodal samples in the instruction-tuning stage. On the other hand, *the latter multi-image scenarios* aim to tackle diverse real-world applications scenarios that involve multi-images. The training data of VPG-C (27) collected 4 new datasets with ChatGPT. Mantis-Instruct (19) compiles existing 11 interleaved datasets and creates 4 new datasets. The proposed M4-Instruct (19) compiles existing 41 interleaved datasets and creates 6 new datasets, covering a much higher scenarios diversity than Mantis-Instruct.

**Interleaved LMMs.** As representative *closed-source LMMs*, both GPT-4V (42) and Gemini (12) support real-world multi-image application scenarios with leading performance. With various public datasets aforementioned, the community has developed *open-source LMMs* equipped with remarkable multi-image proficiency. The ICL performance is typically considered to evaluate multimodal pre-training, which has been adopted in several known LMMs, such as OpenFlamingo (2), IDEFICS series (22; 23), VILA (33) and MM1 (41), Emu2 (51). Otter (25) is initialized from OpenFlamingo, and is fine-tuned on the MIMIC-IT dataset to further improve ICL ability with instruction-tuning. In contrast, the use of instruction-tuning in LMMs for various real-world multi-image applications has been less explored, despite of Mantis (19). The proposed LLaVA-Interleave not only broadens the multi-image scenario itself as demonstrated by the improved experimental results, but also generalize the settings to diverse scenarios with one model, e.g., video, 3D, and single-image. The cross-scenario training leads to emerging capabilities, achieving zero-shot task composition in new multi-image contexts.

**Interleaved Benchmarks.** To assess the interleaved multi-image capabilities of LMMs, there have been several high-quality benchmarks in various scenarios. The ICL benchmarks (20; 49) for LMMs comprehensively evaluate their interleaved skills from few-shot to many-shot settings. For the more challenging multi-image scenarios, previous works mainly focus on a specific domain for evaluation, including NLVR2 (50) for daily-life VQA, MMMU (61) for colleague-level problem-solving, MathVerse-mv (65) and SciVerse-mv (13) for mathematical and scientific reasoning, BLINK (10) to challenge LMMs, and Mantis-Eval (19) for multi-image understanding. To further evaluate LMMs on a collection of multi-image scenarios, DEMON (27) is the first benchmark that compiles dozens of datasets with 477K samples. With the large amount of data and high diversity, DEMON lays a good foundation for multi-image research. Unfortunately, it also inherits a significant amount of low-quality data samples from existing datasets. To facilitate evaluation, the proposed LLaVA-Interleave Bench curate high-quality samples, comprising both specific (synthetic, mathematical, low-level) and general (daily, real-world, text-rich) multi-image scenarios. With 9 newly curated and 13 existing datasets, we categorize them into in-domain (12.9K) and out-domain (4.1K) schemes. Con-current multi-image evaluation benchmarks include MuirBench (54) and ReMI (21).

## 3 INTERLEAVED MULTI-IMAGE TASKS & DATA

### 3.1 TASK OVERVIEW

We observe different computer vision scenarios can be generally represented by the interleaved multi-image format, such as video, 3D, and single-image data. Therefore, to endow LLaVA-Interleave with diverse capabilities, as shown in Figure 2, we adopt the interleaved multi-image format to unify the data input of the following four tasks:

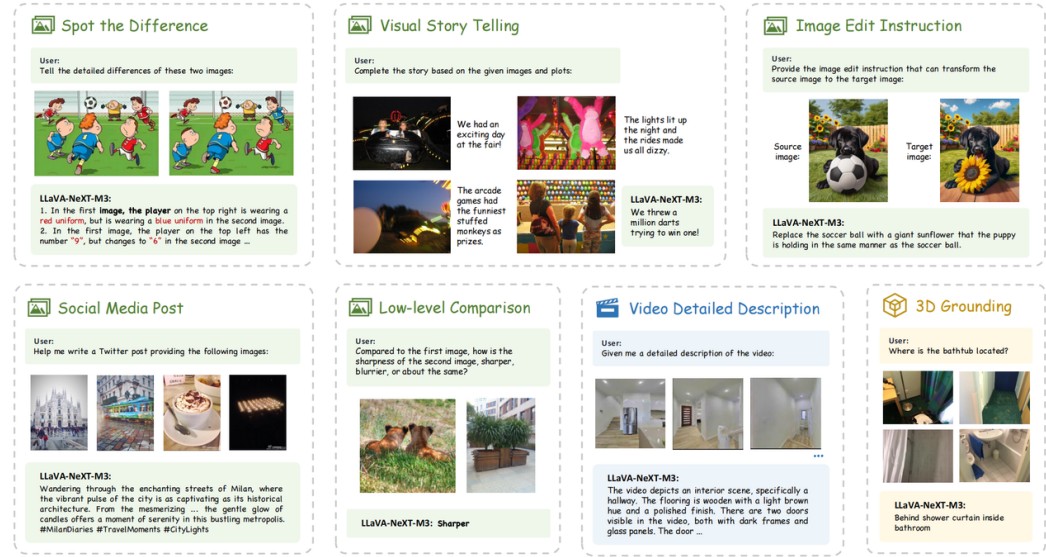

Figure 3: Task examples of M4-Instruct, containing diverse scenarios in multi-image, multi-frame (video), and multi-view (3D).

**Multi-image scenarios** include visual instructions incorporating interleaved vision-language input with multiple images. This setting covers 12 challenging real-world tasks included in our training data, such as spotting the difference, visual story telling, image editing instruction generation, interleaved multi-image dialogue, multi-image puzzle, low-level multi-image assessment, etc.

**Multi-frame scenarios** refer to taking video as input data by sampling it into multiple frames, preserving temporal visual cues across the multi-image sequence. We mainly focus on 2 tasks: video detailed captioning and video VQA.

**Multi-view scenarios** depict 3D environments by multi-view images from different perspectives, where the visual correspondence and disparity can indicate spatial information in the 3D world. For 3D perception, we include 2 tasks: embodied VQA (dialogue and planning), and 3D scene VQA (captioning and grounding).

**Multi-patch scenarios** represent the conventional single-image tasks. With the design of 'any resolution' in LLaVA-NeXT (36), we divide a high-resolution image into multiple low-resolution patches for efficient visual encoding, compatible with our interleaved multi-image format.

## 3.2 M4-INSTRUCT

To empower all-round multi-image capabilities, we meticulously curate a comprehensive training dataset including 1177.6K instances, termed M4-Instruct, widely spanning multi-image, multi-frame, and multi-view scenarios with 14 tasks and 41 datasets, along with multi-patch data to preserve basic single-image performance. We showcase different task examples in Figure 3.

We exhibit a data overview of M4-Instruct in Figure 4, and the detailed data statistics in Table 15. For the multi-image data, most of the datasets are collected from previous public efforts and rigorously converted into our unified format with task-specific instructions, some inspired by DEMON (27) and Mantis (19). On top of that, we also utilize GPT-4V (43) to annotate 3 new tasks to enable more diverse capabilities, i.e., Real-world Difference, Synthetic Difference, and Twitter Post. For the video data, we collect a 255K subset from LLaVA-Hound (63), including 240K video VQA and 15K video detailed captioning. We also include NExT-QA (57) and STAR (55) to expand our video training data. For the 3D data, we widely gather the training set from nuScenes QA (6), ALFRED (48), ScanQA (3), and 3D-LLM (16), covering both outdoor and indoor scenarios. For the single-image

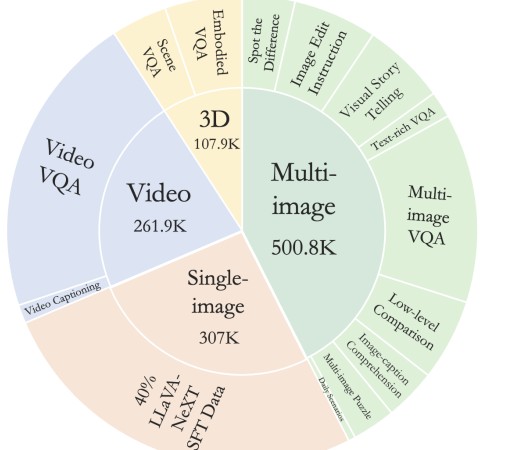 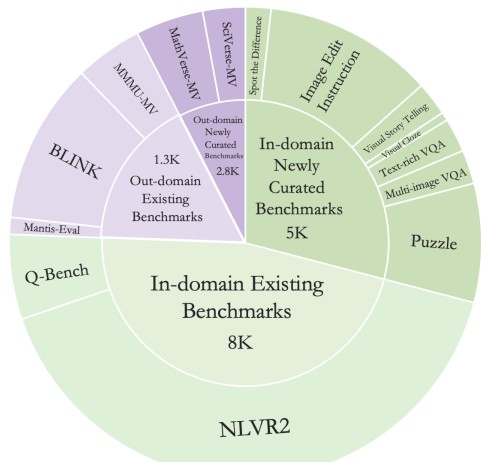

Figure 4: M4-Instruct training data statistics.    Figure 5: LLaVA-Interleave Bench statistics.

data, we randomly sample 40% of the stage-2 fine-tuning data from LLaVA-NeXT (24), which aims to preserve the single-image capacity.

To comprehensively evaluate the interleaved multi-image performance, we introduce the LLaVA-Interleave Bench for LMMs, consisting of 13 challenging tasks with 17K instances. We present a data overview of the benchmark in Figure 3, and the detailed data statistics in Table 16. In detail, we categorize multi-image tasks into two classes:

- *In-domain Evaluation* includes tasks that have been 'seen' during our training, designed to verify the model performance within familiar scenarios. We adopt 5 newly curated multi-image tasks corresponding to training datasets, and 2 existing benchmarks, Q-Bench (56) and NLVR2 (50), with 12.9K in total.

- *Out-domain Evaluation* involves tasks that don't overlap with training scenarios, aiming to reveal the generalization capacity of LMMs. We construct 2 new tasks for multi-image mathematical (MathVerse (65)) and scientific (SciVerse (13)) comprehension, and utilize 3 existing benchmarks, Mantis-Eval (19), BLINK (10), and MMMU (60), with 4.1K in total.

## 4 INTERLEAVED VISUAL INSTRUCTION TUNING

In this section, we introduce several key techniques during the interleaved visual instruction tuning of LLaVA-Interleave. For architecture designs, we follow LLaVA-NeXT (24) to adopt the most general framework, i.e., a vision encoder (62), an intermediate projector, and an LLM (4). Then, we consider the following techniques to achieve improved multi-image performance.

**Technique 1: Continue training from single-image models.** The interleaved multi-image tasks can be regarded as an extension of single-image scenarios, more flexible in formats and challenging in reasoning. Therefore, to better leverage the pre-trained single-image proficiency, we adopt an off-the-shelf LLaVA-NeXT-Image (24) as the base model, which has gone through a stage-1 image-caption pre-training and a stage-2 single-image fine-tuning. On top of this model, we perform the interleaved multi-image instruction tuning with our M4-Instruct dataset.

**Technique 2: Mixed Interleaved data formats during training.** We adopt two format choices for the positions of image tokens during the interleaved multi-image training. The first is to place all the image tokens in front of the prompt, while maintaining the placeholders ⟨image⟩ within the text, denoted as 'In-the-front format'. The second preserves the interleaved format to put image tokens in the place they are originally in, i.e., the positions of ⟨image⟩, denoted as 'interleaved format'. In this way, LLaVA-Interleave supports more flexible inference modes, exhibiting robustness to different input formats.

Table 1: Results on our LLaVA-Interleave Bench. SD: Spot the Difference, IE: Image Edit Instruction, VST: Visual Story Telling, TRVQA: Text-rich VQA, MIVQA: Multi-image VQA, QB: Q-Bench, SQ: ScanQA, Math: MathVerse-mv, Sci: SciVerse-mv.

| Model | In-domain Evaluation | | | | | | | | | Out-domain Evaluation | | | | | |
|---|---|---|---|---|---|---|---|---|---|---|---|---|---|---|---|
| | Avg | SD | IE | VST | TRVQA | MIVQA | Puzzle | QB | NLVR2 | Avg | Math | Sci | Mantis | BLINK | MMMU-mv |
| GPT-4V (43) | 39.2 | 12.5 | 11.0 | 10.9 | 54.5 | 52.0 | 17.1 | 76.5 | 88.8 | 57.8 | 60.3 | 66.9 | 62.7 | 51.1 | 47.9 |
| LLaVA-NeXT-Image (7B) (36) | 32.4 | 12.9 | 13.2 | 10.1 | 59.6 | 39.4 | 9.0 | 51.0 | 68.0 | 29.4 | 13.5 | 12.2 | 46.1 | 41.8 | 33.5 |
| VPG-C (7B) (28) | 35.8 | 27.8 | 15.2 | 21.5 | 38.9 | 46.8 | 2.4 | 57.6 | 73.2 | 34.5 | 24.3 | 23.1 | 52.4 | 43.1 | 29.4 |
| Mantis (7B) (19) | 39.6 | 17.6 | 11.2 | 12.5 | 45.2 | 52.5 | 25.7 | 69.9 | 87.4 | 39.3 | 27.2 | 29.3 | 59.5 | 46.4 | 34.1 |
| LLaVA-Interleave | | | | | | | | | | | | | | | |
| 0.5B | 43.9 | 34.3 | 21.6 | 29.7 | 63.9 | 54.8 | 35.4 | 52.0 | 67.8 | 33.1 | 13.3 | 12.2 | 45.6 | 39.2 | 28.6 |
| 7B | 58.6 | 37.1 | 24.3 | 33.1 | 76.1 | 87.5 | 48.7 | 74.2 | 88.8 | 42.8 | 32.8 | 31.6 | 62.7 | 52.6 | 34.5 |
| 14B | **62.3** | **40.5** | **24.5** | **33.3** | **78.6** | **95.0** | **59.9** | **76.7** | **91.1** | **44.3** | **33.4** | **32.7** | **66.4** | **52.1** | **37.1** |

**Technique 3: Combining different data scenarios improves individual task performance.** Most existing works conduct supervised fine-tuning with only one type of data source, e.g., multi-image tuning of Mantis (19) and multi-frame tuning of LLaMA-VID (31). Instead, we utilize the M4-Instruct to simultaneously conduct instruction tuning with four different tasks (multi-image/frame/view/patch). With a unified interleaved format, distinct data scenarios have the potential to provide complementary semantics and instruction-following capabilities.

## 5 EXPERIMENTS

In Section 5.1, we first introduce our evaluation schemes and implementation details. Then, in Section 5.2, we report and analyze the quantitative results in four interleaved multi-image scenarios.

### 5.1 SETTINGS

**Evaluation Schemes.** We evaluate LLaVA-Interleave on four real-world interleaved scenarios, i.e., multi-image, multi-frame (video), multi-view (3D), and multi-patch (single-image).

- *For multi-image evaluation*, we adopt the proposed LLaVA-Interleave Bench covering comprehensive in-domain and out-domain tasks.

- *For video evaluation*, we utilize the existing NExT-QA (57), MVBench (30), Video Detailed Description (VDD) (67), and ActivityNet-QA (Act) (59). For ActivityNet-QA, we present both the accuracy and GPT score (Acc/Score). We also evaluate on VideoChat-GPT (VCG) (39) with five metrics: CI (Correctness of Information), DO (Detail Orientation), CU (Context Understanding), TU (Temporal Understanding), and CO (Consistency).

- *For 3D evaluation*, we select ScanQA (3), two tasks from 3D-LLM (16), i.e., 3D-assisted Dialogue and Task Decomposition, and also curate two new test set from nuScenes VQA (6) and ALFRED (48).

**Implementation Details.** Following the same architecture in LLaVA-NeXT (24), our LLaVA-Interleave adopts Qwen 1.5 (5) as the base LLM with 0.5B, 7B and 14B parameters, SigLIP-400M (62) ($384 \times 384$) as the vision encoder, and a two-layer MLP as the projection layer.

### 5.2 MAIN RESULTS

**Multi-image Results.** As reported in Table 1, the average multi-image performance of LLaVA-Interleave surpasses previous open-source models in both in- and out-domain benchmarks. For in-domain evaluation, our model demonstrates significant advantages across various tasks as expected, due to the multi-image instruction tuning with M4-Instruct. For out-domain evaluation, LLaVA-Interleave also showcases superior generalization capacity within novel scenarios, e.g., comparable to GPT-4V on Mantis-Eval and BLINK.

**Multi-frame (Video) Results.** Compared with previous video-based LMMs under similar model sizes, LLaVA-Interleave achieves superior results on many benchmarks in Table 2, though not

Table 2: Results on multi-frame (video) benchmarks. VDD: Video Detailed Description. CI (Correctness of Information), DO (Detail Orientation), CU (Context Understanding), TU (Temporal Understanding), and CO (Consistency).

| Model | NExTQA | MVBench | ActivityNet-QA | VDD | VideoChat-GPT | | | | |
|---|---|---|---|---|---|---|---|---|---|
| | | | | | CI | DO | CU | TU | CO |
| GPT-4V (43) | - | - | - | 4.00 | 4.09 | 3.88 | 4.37 | 3.94 | 4.02 |
| VideoChatGPT (7B) (40) | - | - | 35.2/2.70 | - | 2.40 | 2.52 | 2.62 | 1.98 | 2.37 |
| Video-LLaVA (7B) (32) | - | - | 45.3/3.30 | - | 2.87 | 2.94 | 3.44 | 2.45 | 2.51 |
| VISTA-LLaMA (7B) (38) | - | - | 48.3/3.30 | - | 2.44 | 2.31 | 2.64 | 3.18 | 2.26 |
| VideoChat2 (7B) (29) | 68.6 | 51.9 | 49.1/3.30 | - | 3.02 | 2.88 | 3.51 | 2.66 | 2.81 |
| LLaMA-VID (7B) (31) | - | 50.2 | 47.4/3.30 | 2.84 | 3.01 | 2.97 | 3.54 | 2.53 | 2.60 |
| LLaVA-NeXT-Video (7B) (67) | - | - | 53.5/3.20 | 3.32 | 3.39 | 3.29 | 3.92 | 2.60 | 3.12 |
| LLaVA-NeXT-Video-DPO (7B) | - | - | 60.2/3.50 | 3.72 | 3.64 | 3.45 | 4.17 | 2.95 | 4.08 |
| LLaVA-NeXT-Video-DPO (34B) | - | - | **64.4/3.60** | 3.84 | 3.81 | 3.55 | 4.24 | 3.14 | 4.12 |
| LLaVA-Interleave | | | | | | | | | |
| 0.5B | 59.5 | 45.6 | 48.0/2.84 | 3.25 | 3.12 | 2.97 | 3.62 | 2.36 | 3.27 |
| 7B | 78.2 | 53.1 | 55.3/3.13 | 3.57 | 3.51 | 3.28 | 3.89 | 2.77 | 3.68 |
| 14B | 79.1 | **54.9** | 56.2/3.19 | 3.59 | 3.65 | 3.37 | 3.98 | 2.74 | 3.67 |
| 7B (DPO) | **77.9** | 52.3 | 55.0/3.13 | **3.90** | **3.99** | **3.61** | **4.24** | **3.19** | **4.12** |

Table 3: Results on multi-view (3D) benchmarks. 3D-assisted Dialogue and Task Decomposition are evaluation tasks from 3D-LLM.

| Model | Avg | 3D-assisted Dialogue | Task Decomposition | ScanQA (val) | ALFRED | nuScenes VQA |
|---|---|---|---|---|---|---|
| Flamingo (1) | 20.5 | 27.9 | 33.2 | 31.1 | 5.3 | 4.9 |
| GPT-4V (43) | 34.6 | 31.2 | 35.4 | 32.6 | 10.3 | 63.7 |
| Point-Bind & LLM (14) | 22.5 | 38.3 | 35.8 | 34.6 | 0.6 | 3.3 |
| 3D-LLM (17) | 22.9 | 39.3 | 37.8 | 35.7 | 1.4 | 0.4 |
| Mantis (7B) (19) | 18.7 | 2.60 | 14.7 | 16.1 | 14.0 | 46.2 |
| LLaVA-Interleave | | | | | | |
| 0.5B | 53.0 | 67.2 | 48.5 | 29.3 | 57.0 | 62.8 |
| 7B | 58.2 | 69.3 | 51.4 | 32.2 | 61.6 | 76.5 |
| 14B | **59.2** | **70.6** | **52.2** | **34.5** | **62.0** | **76.7** |

specifically designed for video tasks. We also follow LLaVA-Hound to add DPO training after our M4-Instruct tuning. After adding DPO, our 7B model attains SoTA performance on VDD and VideoChat-GPT benchmarks, surpassing the previous LLaVA-NeXT-Video (34B). This demonstrates the effective temporal understanding and reasoning capabilities of our model across sequential frames. Note that we calculate the average scores by multiplying a weight of 10 times by the score of Video Detailed Description and VideoChat-GPT.

**Multi-view (3D) Results.** For 3D perception in Table 3, our model also obtains leading results for both indoor and outdoor scenarios on five in-domain benchmarks. Compared to 3D-LLM and Point-LLM with additional point clouds as input, LLaVA-Interleave only accepts multi-view images to interpret the 3D world, attaining significantly higher scores in challenging 3D scenarios.

**Multi-patch (single-image) Results.** We also add 307k (40%) of original LLaVA-NeXT single-image data, which makes our model capable of doing single-image tasks. We use the *anyres* training for single-image data, which divides an image into multiple patches, forming another multi-image setting. As shown in Table 5, we maintain the single-image performance of LLaVA-NeXT-Image. As single-image data is of high quality and diversity, adding single-image data also improves the instruction-following ability and enables task transfer from single-image to multi-image, which is demonstrated in Section 6.

Table 5: Results on multi-patch (single-image) benchmarks with different LLM sizes. 'Single' and 'Interleave' denote LLaVA-NeXT-Image and our model, respectively.

| Model | LLM | Avg | AI2D | ChartQA | DocVQA | MME | SciQA | POPE |
|-------|-----|-----|------|---------|--------|-----|-------|------|
| Single | 0.5B | 59.8 | 51.7 | 50.2 | 59.1 | 52.8 | 60.0 | 85.4 |
| Interleave | | **60.5** | 52.2 | 52.2 | 59.2 | 52.0 | 60.6 | 86.8 |
| Single | 7B | 72.3 | 72.7 | 66.3 | 75.6 | 61.0 | 71.1 | 86.9 |
| Interleave | | **73.3** | 73.9 | 67.2 | 75.7 | 63.5 | 72.6 | 86.8 |
| Single | 14B | **77.2** | 77.5 | 72.1 | 80.0 | 67.7 | 78.9 | 87.3 |
| Interleave | | 76.4 | 76.5 | 71.2 | 78.9 | 66.2 | 77.4 | 87.9 |

Table 6: Ablation on whether to continue training from single-image models. QB: Q-Bench, Act: ActivityNet-QA, MVB: MVBench, VDD: Video Detailed Description, MME*: Throughout our paper, we convert MME's score to accuracy by summing up the perception and cognition scores and dividing 2800, SQA: Scienceqa-IMG.

| Continue training | Multi-image | | | | Multi-frame | | | Multi-view | Single-image | | | | | |
|-------------------|--------|-------|------|-------|----------|------|------|--------|------|---------|--------|------|------|------|
| | Mantis | BLINK | QB | NLVR2 | Act | MVB | VDD | ScanQA | AI2D | ChartQA | DocVQA | MME* | POPE | SQA |
| From stage-1 pre-training | 41.0 | 37.6 | 47.0 | 54.0 | 44.7/2.17 | 43.0 | 2.96 | 27.7 | 46.3 | 38.3 | 47.5 | 47.1 | 85.4 | 59.4 |
| From single-image models | 45.6 | 39.2 | 52.0 | 67.8 | 48.0/2.84 | 45.6 | 3.25 | 29.3 | 52.2 | 52.2 | 59.2 | 52.0 | 86.8 | 60.6 |

## 5.3 ABLATIONS OF PROPOSED TECHNIQUES

We study the effectiveness of the three proposed training techniques in Section 4 as below.

**In Table 6**, we compare training strategies. It is seen that initialization from a good single-image model checkpoint (from Stage-2) can consistently enhance the interleaved multi-image performance, than directly from a Stage-1 model checkpoint.

**In Table 7**, our mixed-format training can benefit the results of both two input formats.

**In Table 8**, we progressively incorporate single-image and multi-image data upon the video data. The integration of more sources contributes to enhanced performance, compared with models from individual visual scenarios.

## 6 EMERGING CAPABILITIES

In this section, we show some example to demonstrate the emerging capabilities of our model. Emerging capabilities means the capabilities do not trained during training but demonstrated when inference. We mainly showcase the emerging capabilities from three aspects:

1. **Task Transfer from Single-image to Multi-image:** The capability to reason over one image and tell the funny part is initially observed in single-image models (35), and not included in our multi-image training. As shown in Table 9, our model is capable of *analyzing the fun part within multiple images*. This new task is probably emerged by the composition of the single-image capability and multi-image VQA training.

2. **Task Transfer from Image to Video:** We only include the multi-image Twitter post task in the M4-Instruct training, while our model can directly perform the *witter post from a video*, as shown in Table 10. This new task is probably composed by the training data of multi-image Twitter post and video VQA tasks.

3. **Real-world Applications:** In Tables 11 and 12, we showcase one real-world scenario that is not explicitly contained in our interleaved training data, which are multi-image painting style recognition and PPT summary & QA. This demonstrates our generalization potentials to a broader spectrum of applications. More examples are in Appendix.

Table 7: Ablation on mixed interleaved data formats. We select several important tasks within our LLaVA-Interleave Bench for the ablation study.

| Training Setting | Inference Setting | Avg | Spot the Difference | Visual Story Telling | Text-rich VQA | Q-Bench |
|---|---|---|---|---|---|---|
| In-the-front | Interleaved | 52.9 | 36.8 | 30.5 | 70.1 | 74.0 |
| | In-the-front | 54.3 | 36.6 | 32.8 | 74.7 | 75.3 |
| Interleaved | Interleaved | 55.4 | 37.8 | 32.9 | 76.2 | 76.0 |
| | In-the-front | 52.4 | 36.1 | 29.0 | 72.9 | 71.8 |
| Mixed | Interleaved | 57.0 | 38.3 | 32.5 | 78.1 | 76.9 |
| | In-the-front | 56.6 | 37.9 | 32.5 | 78.4 | 76.3 |

Table 8: Ablation on the improvement of combined data scenarios for video tasks. CI (Correctness of Information), DO (Detail Orientation), CU (Context Understanding), TU (Temporal Understanding), and CO (Consistency).

| Data | NExT-QA | VDD | VideoChatGPT | | | | |
|---|---|---|---|---|---|---|---|
| | | | CI | DO | CU | TU | CO |
| Video | 42.6 | 3.46 | 3.47 | 3.27 | 3.87 | 2.74 | 3.61 |
| Video + Single-image | 67.7 | 3.49 | 3.46 | 3.30 | 3.85 | 2.71 | 3.60 |
| Video + Multi-image | 77.7 | 3.50 | 3.50 | 3.31 | 3.90 | 2.70 | 3.63 |
| Video + Both | 78.2 | 3.58 | 3.50 | 3.27 | 3.87 | 2.77 | 3.68 |

Table 9: An example of emerging capability: multi-image fun part analyze.

---

**Task Transfer from Single-image to Multi-image**

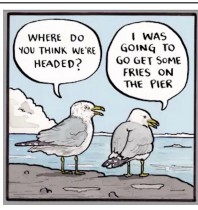 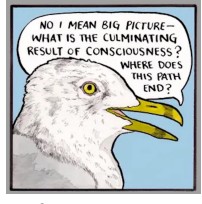 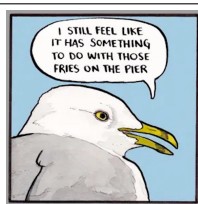

| User | What is fun about the images? |
|---|---|
| Ours | The humor in these images comes from the anthropomorphized birds engaging in human-like activities and conversations. In the first image, one bird is asking about going to get some fries on the pier, which is a human activity. The second image shows a bird asking about the "culminating result of consciousness," which is a philosophical question rather than a literal one. The third image has a bird expressing its desire to do something with the fries on the pier, again referencing a human activity. The anthropomorphism of the birds adds a layer of humor by attributing human thoughts and desires to animals. |

---

# 7 CONCLUSION

In conclusion, our research highlights the transformative potential of LLaVA-Interleave in unifying and advancing the capabilities of Large Multimodal Models (LMMs) across diverse visual tasks. By leveraging the interleaved data format, we effectively integrate multi-image, video, 3D, and single-image scenarios, offering a cohesive approach to handling thwoese varied challenges. The introduction of the comprehensive **M4-Instruct** dataset and the **LLaVA-Interleave Bench** provides a solid foundation for training and evaluating LMMs, ensuring high-quality performance across multiple domains. Our extensive experiments validate that LLaVA-Interleave not only sets new state-of-the-art benchmarks in multi-image tasks but also maintains exceptional performance in single-image tasks. Furthermore, the model exhibits promising emerging capabilities, such as cross-task transfer, showcasing its versatility and potential for broader applications. This work sets a new precedent in the field, paving the way for future advancements in multimodal AI and complex visual understanding tasks.

Table 10: An example of emerging capability: Twitter post for videos.

| **Task Transfer from Image to Video** |
| --- |
| 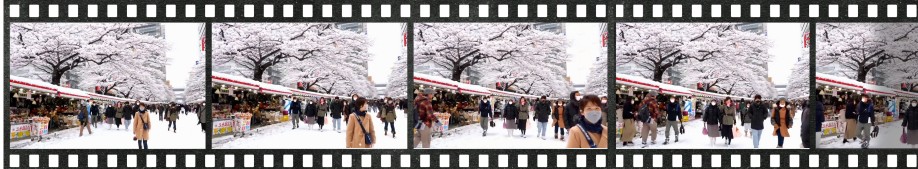 |
| User | Help me write a twitter post to describe this video. |
| Ours | "Feeling the chill in the air, but the cherry blossoms are a sight to behold! Walking down the street, each person bundled up against the winter chill. The snow-covered trees and market stalls add to the serene beauty of this winter wonderland. #CherryBlossoms #WinterWalk #CityScenes". |

Table 11: An example of emerging capability: recognizing artists' painting styles.

| **Real-World Applications** |
| --- |
| 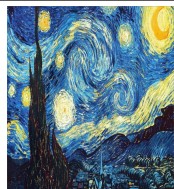 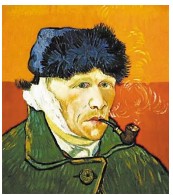 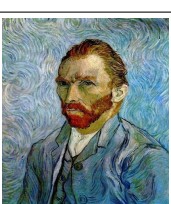 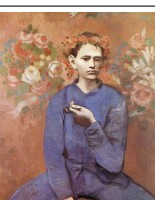 |
| User | Do you kown who draw these paintings? |
| Ours | Yes, these paintings are by the renowned artist Vincent van Gogh. The first painting is his famous work "The Starry Night," which depicts a swirling night sky with a bright crescent moon and stars. The second painting is his self-portrait, which shows him smoking a pipe and wearing a hat. The third painting is also a self-portrait, but it features a different expression and hairstyle. The fourth painting is by another artist, not Van Gogh, and depicts a person seated with flowers in the background. |

Table 12: An example of emerging capability: PPT summary & VQA.

| **Real-World Applications** |
| --- |
| 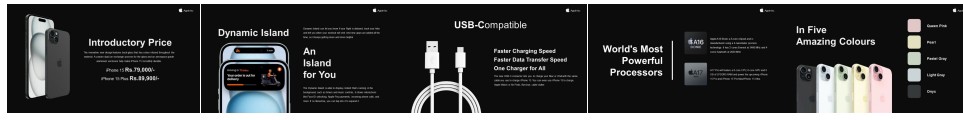 |
| User | The images are the PPT of iPhone 15 review. can you summarize the main information? |
| Ours | The PPT features a sleek design with a black background and includes text and images related to the iPhone 15. It highlights the phone's innovative features such as a custom dual glass back cover, a dynamic island display for delivery tracking, and a USB-C compatible fast charging cable. The color options are showcased in a gradient of pink, green, gray, and onyx. The presentation also emphasizes the phone's powerful processors, including the A16 Bionic chip and A17 Pro chip. |
| User | What is the price of iPhone 15 plus? |
| Ours | The price of iPhone 15 plus is Rs. 89,000 as mentioned in the PPT slide. |
| User | Why the dynamic island useful? |
| Ours | The dynamic island is useful for delivery tracking, as it allows the user to know when their order will arrive and if there are any delays. It also provides a new way to interact with the phone's interface, making it easier to navigate and access various features. |

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

# A   DATA STATISTICS

The detailed data statistics of M4-Instruct is shown in Table 15.

The detailed data statistics of LLaVA-Interleave Bench is shown in Table 16.

# B   ABLATION STUDY

## B.1   POOL VS NOT POOL VISION TOKENS FOR VIDEO TASKS.

Similar to LLaVA-NEXT-Video, we adopt a "Pooling to 1/4" strategy for which we pool the width and heighs of feature maps to 1/2 therefore reducing the number to totals to 1/4. We study the impact of image token pooling. We train and infer our model under two settings: pooling to 1/4 and not pooling with ShareGPTVideo-Caption+QA(255K) data. Pooling to a 1/4 setting is similar to LLaVA-NEXT-Video, which uses the pooling technique to trade-off between the number of image tokens and the number of frames. In our experiment, we find that not pooling yields better performance under similar #image tokens. During training, we sample 10 frames for videos. In this table, we also observe that adding more frames (from 10 to 16) during inference improves performance.

## B.2   IMPACT OF VIDEO DPO TRAINING ON OTHER TASKS.

In Table 14, we compare the results of doing video DPO on other tasks. Though DPO significantly improves the video performance as shown in Table 2, it slightly impacts the performance of other tasks.

Table 13: Ablation to compare pooling and not pooling.

| Training | Inference | #frames | # Image tokens | Act | Avg | VDD | VideoChatGPT | | | | |
|---|---|---|---|---|---|---|---|---|---|---|---|
| | | | | | | | CI | DO | CU | TU | CO |
| Pooling 1/4 | Pooling 1/4 | 40 | 40x729x1/4=10x729 | 52.8/3.53 | 3.35 | 3.38 | 3.46 | 3.25 | 3.87 | 2.59 | 3.57 |
| Pooling 1/4 | Pooling 1/4 | 64 | 64x729x1/4=16x729 | 52.7/3.53 | 3.33 | 3.38 | 3.45 | 3.23 | 3.86 | 2.49 | 3.55 |
| Not Pooling | Not Pooling | 10 | 10x729 | 52.9/3.48 | 3.38 | 3.46 | 3.43 | 3.26 | 3.85 | 2.64 | 3.61 |
| Not Pooling | Not Pooling | 16 | 16x729 | **54.4/3.51** | **3.41** | **3.46** | **3.48** | **3.28** | **3.87** | **2.74** | **3.62** |

Table 14: Ablation on the impact of video dpo on the performance of other tasks. QB: Q-Bench, Act: ActivityNet-QA, MVB: MVBench, VDD: Video Detailed Description, MME*: Throughout our paper, we convert MME's score to accuracy by summing up the perception and cognition scores and dividing 2800, SQA: Scienceqa-IMG.

| Setting | Multi-image | | | | Multi-view | Single-image | | | | | | |
|---|---|---|---|---|---|---|---|---|---|---|---|---|
| | Mantis | BLINK | QB | NLVR2 | ScanQA | AI2D | ChartQA | DocVQA | MME* | POPE | SQA |
| Before Video-DPO | 62.7 | 52.7 | 73 | 89.1 | 32.2 | 73.9 | 67.2 | 75.7 | 63.5 | 85.4 | 72.6 |
| After Video-DPO | 60.8 | 51.7 | 86.8 | 87.7 | 25.5 | 72.2 | 56.1 | 73.1 | 62.6 | 86.6 | 71.7 |

Table 15: M4-Instruct detailed datasets.

| Task | Dataset | Scenario | # Samples |
|---|---|---|---|
| **Multi-image Scenarios** | | | |
| Spot the Difference(42.6K) | Real-world Difference | Realistic | 6.7K |
| | Synthetic Difference | Synthetic | 7.0K |
| | Spot-the-Diff | Surveilance | 10.8K |
| | Birds-to-Words | Birds | 14.2K |
| | CLEVR-Change | Solids | 3.9K |
| Image Edit Instruction(67.7K) | HQ-Edit | Synthetic | 50K |
| | MagicBrush | Realistic | 14.2K |
| | IEdit | Realistic | 3.5K |
| Visual Story Telling(67.5K) | AESOP | Cartoon | 6.9K |
| | FlintstonesSV | Cartoon | 22.3K |
| | PororoSV | Cartoon | 12.3K |
| | VIST | Realistic | 26K |
| Text-rich VQA(21.3K) | WebQA | Webpage | 9.3K |
| | TQA | Textbook | 8.2K |
| | OCR-VQA | OCR | 1.9K |
| | DocVQA | Document | 1.9K |
| Multi-image VQA(153.5K) | NLVR2 | Realistic | 86.4K |
| | MIT-States_StateCoherence | General | 1.9K |
| | MIT-States_PropertyCoherence | General | 1.9K |
| | RecipeQA_ImageCoherence | Recipe | 8.7K |
| | VISION | Industrial | 9.9K |
| | Multi-VQA | General | 5K |
| | IconQA | General | 34.6K |
| Low-level Comparison(65.9K) | Coinstruct | Low-level | 50K |
| | Dreamsim | Low-level | 15.9K |
| Image-caption Comprehension (41.8K) | ImageCoDe | General | 16.6K |
| | Contrast-Caption | General | 25.2K |
| Daily Scenarios (5.7K) | MMChat_Twitter_Post | General | 5.7K |
| Multi-image Puzzle (35K) | Raven | Abstract | 35K |
| **Multi-frame (Video) Scenarios** | | | |
| Video QA(246.9K) | NExT-QA | General | 3.9K |
| | STAR | General | 3K |
| | ShareGPTVideo-VQA | General | 240K |
| Video Detailed Captioning (15K) | ShareGPTVideo-Caption | General | 15K |
| **Multi-view (3D) Scenarios** | | | |
| Scene VQA(45.4K) | Nuscenes | Outdoor | 9.8K |
| | ScanQA | Indoor Realistic | 25.6k |
| | 3D-LLM-Scene | Indoor Realistic | 10K |
| Embodied VQA(62.5K) | ALFRED | Indoor Synthetic | 22.6K |
| | 3D-LLM-Dialogue | Indoor Realistic | 20K |
| | 3D-LLM-Planning | Indoor Realistic | 19.9K |
| **Single-image Scenarios** | | | |
| Single-image Tasks(307K) | Randomly sampling 40% SFT data of LLaVA-NeXT | General | 307K |

Table 16: LLaVA-Interleave Bench detailed datasets.

| Task | Dataset | Scenario | # Samples |
|---|---|---|---|
| **In-domain Evaluation - Newly Curated Benchmarks** | | | |
| Spot the Difference(0.3K) | Spot-the-Diff | Surveilance | 0.1K |
| | Birds-to-Words | Birds | 0.1K |
| | CLEVR-Change | Solids | 0.1K |
| Image Edit Instruction(2K) | HQ-Edit | Sythentic | 1K |
| | MagicBrush | Realistic | 0.9K |
| | IEdit | Realistic | 0.1K |
| Visual Story Telling(0.4K) | AESOP | Cartoon | 0.1K |
| | FlintstonesSV | Cartoon | 0.1K |
| | PororoSV | Cartoon | 0.1K |
| | VIST | Realistic | 0.1K |
| Text-rich VQA(0.4K) | WebQA | Webpage | 0.1K |
| | TQA | Textbook | 0.1K |
| | OCR-VQA | OCR | 0.1K |
| | DocVQA | Document | 0.1K |
| Multi-image VQA(0.4K) | MIT-States_StateCoherence | General | 0.1K |
| | MIT-States_PropertyCoherence | General | 0.1K |
| | RecipeQA_ImageCoherence | Recipe | 0.1K |
| | VISION | Industrial | 0.1K |
| Puzzle (1.4K) | Raven | Abstract | 1.4K |
| **In-domain Evaluation - Existing Benchmarks** | | | |
| NLVR2 (7K) | NLVR2 | Realistic | 7K |
| Q-Bench (1K) | Q-Bench | Low-level | 1K |
| **Out-domain Evaluation - Newly Curated Benchmarks** | | | |
| MathVerse-mv (0.8K) | MathVerse (Vision Dominant) | Math Diagram | 0.8K |
| SciVerse-mv (0.4K) | SciVerse (Vision Dominant) | Scientific Diagram | 0.4K |
| **Out-domain Evaluation - Existing Benchmarks** | | | |
| Mantis-Eval (0.2K) | Mantis-Eval | General | 0.2K |
| BLINK (1.9K) | BLINK | General | 1.9k |
| MMMU-mv (test) (0.8K) | MMMU | Scientific Diagram | 0.8K |

## C MORE REAL-WORLD APPLICATIONS

Table 17: An example of emerging capability: multi-doc VQA.

**Real-World Applications**

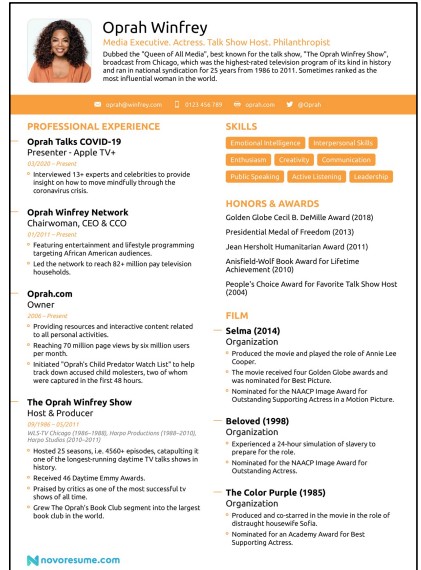 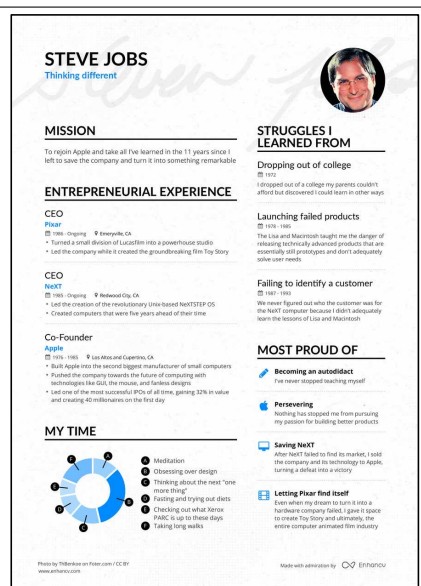

| | |
|---|---|
| User | Hi, there are two candidates, can you provide a brief description for each of them for me? |
| Ours | Certainly! Here's a brief description of each candidate: 1. Oprah Winfrey: - Professional Experience: Known for her talk show, "The Oprah Winfrey Show," and her work on "COVID-19" as a presenter on Apple TV. - Skills: Emphasized on emotional intelligence, interpersonal skills, and creativity. - Honors & Awards: Notable achievements include the Golden Globe Award for Best Television Series and the Golden Globe Award for Best Television Special. 2. Steve Jobs: - Professional Experience: Co-founder of Apple Inc., where he led the development of iconic products like the iMac, iPod, and iPhone. - Skills: Highlighted as a thinker with a focus on innovation and problem-solving. - Honors & Awards: Recognized with numerous accolades, including the MacArthur Fellowship and the NeXT Computer Science Award. Please note that these descriptions are based on the information provided in the images and do not include any personal details or specific accomplishments mentioned in the images. |
| User | Which of them better suits the role of the Congressman of the US? Why? |
| Ours | The first resume, featuring Oprah Winfrey, would better suit the role of a Congressman because it highlights skills and experiences that are relevant to political office, such as leadership, public speaking, and media experience. These qualities are essential for a politician to effectively represent their constituents and engage in legislative processes. The second resume, while it could be suitable for a creative or entrepreneurial role, does not specifically mention skills or experiences that align with the responsibilities of a Congressman. |

