# OpenReview forum: "LLaVA-Interleave: Tackling Multi-image, Video, and 3D in Large Multimodal Models"
_ICLR.cc/2025/Conference — ICLR 2025 Spotlight_

### Official Review · Reviewer_jqfW · 2024-10-27

**Soundness:** 4
**Presentation:** 3
**Contribution:** 4
**Rating:** 8
**Confidence:** 4

**Summary:**

This paper introduces LLaVA-NeXT-Interleave, a large multimodal model (LMM) designed to unify and enhance capabilities across multi-image, video, and 3D tasks while maintaining performance on single-image tasks. The authors propose using an interleaved image-text data format as a universal template to represent different computer vision scenarios. They compile a comprehensive dataset called M4-Instruct, consisting of over 1 million samples spanning four primary domains with 14 tasks and 41 datasets. Additionally, they create the LLaVA-Interleave Bench, a diverse set of benchmarks for evaluating multi-image performance. Extensive experiments demonstrate that LLaVA-NeXT-Interleave achieves state-of-the-art results across various benchmarks and exhibits emerging capabilities such as cross-task transfer.

**Strengths:**

Innovative Unification Framework: The paper presents a novel approach by using an interleaved image-text format to unify multi-image, video, and 3D tasks within a single LMM framework. This unification is both practical and innovative, addressing a significant gap in the current research.

Comprehensive Dataset and Benchmark Creation: The compilation of the M4-Instruct dataset and the LLaVA-Interleave Bench provides valuable resources for the community. These datasets are extensive and cover a wide range of tasks and scenarios, enhancing the reproducibility and applicability of the research.

State-of-the-Art Performance: The proposed model achieves leading results across various benchmarks in multi-image, video, and 3D tasks while maintaining strong performance on single-image tasks. This demonstrates the effectiveness of the approach.

Emerging Capabilities: The model exhibits emerging capabilities such as cross-task transfer and zero-shot task composition, indicating strong generalization and adaptability to new scenarios.

Detailed Experimental Analysis: The paper includes extensive experiments and ablation studies that thoroughly evaluate the model's performance and validate the proposed techniques.

**Weaknesses:**

Limited Details on Data Curation: While the paper introduces a large and diverse dataset, it provides limited information on the data curation process, quality control, and potential biases. More details on how data quality and diversity were ensured would strengthen the work.

Computational Efficiency Concerns: The paper does not thoroughly discuss the computational efficiency or resource requirements of the proposed model, especially when handling multiple modalities. Comparisons in terms of model complexity and inference speed with existing models would be beneficial.

Scope of Evaluation Metrics: The evaluation focuses mainly on quantitative performance metrics. Incorporating qualitative analyses or user studies could provide additional insights into the model's real-world applicability and limitations.

Scalability and Practical Deployment: Potential challenges related to scaling the model to larger datasets or deploying it in practical applications are not fully explored. Discussion on these aspects would enhance the paper's impact.

**Questions:**

Data Curation Process: Can the authors provide more details on the data collection and curation process for the M4-Instruct dataset? Specifically, how were data quality and diversity ensured, and what measures were taken to mitigate potential biases?

Computational Resources: How does LLaVA-NeXT-Interleave perform in terms of computational efficiency compared to other state-of-the-art models? Are there any optimizations implemented to handle the increased complexity of processing multiple modalities?

Scalability: Have the authors explored the scalability of the model when trained on even larger datasets or when applied to more complex tasks? Are there any observed limitations or degradation in performance?

Generalization to Unseen Tasks: While the model demonstrates emerging capabilities, how well does it generalize to completely unseen tasks or modalities not included in the training data?

---

> ### Author Response · Authors · 2024-11-20
> **Response to the reviewer [Concern 1]**
>
> We genuinely appreciate your insightful comments and recognition of our work. We have carefully addressed your feedback with detailed responses and have updated the relevant sections in the revised manuscript to address your concerns.
>
> **Details on data curation, quality control, and potential bias.**
>
> ----
>
> We apologize for any missing details in the description of our data construction process. In the M4-Instruct dataset, we not only aggregated existing datasets but also created new ones through innovative methods, such as leveraging GPT-4 Vision (GPT-4-V) prompts. For instance, notable contributions is the construction of the MMChat_Twitter_Post, ScanQA, and MagicBrush dataset.
>
> - **For the MMChat_Twitter_Post dataset**, we utilized images sourced from an existing social media dataset MMChat. These images were input into GPT-4-V, paired with a language prompt specifically designed to generate Twitter-like posts. This process allowed us to create realistic and contextually relevant data entries, enhancing the overall dataset quality.
>
> - **For the ScanQA dataset**, since the original ScanQA only includes paired question-answer data with 3D point clouds, we sourced the previous ScanNet dataset and identified the mapping relationship between each point cloud and its corresponding video. From each video, we uniformly sampled 16 frames and prompt GPT-4-V to refine the aligned question-answer pairs in ScanQA, constructing multi-image instructions.
>
> - **For the MagicBrush dataset**, we transformed its original image editing task into various conversational formats. In general, we concatenate the source and target image in the instruction as the condition and set the task as predicting the corresponding editing prompt. For cases where multiple edits are required between two images, we either structure the data into multi-turn conversations, enabling models to progressively interpret the differences, or prompt GPT-4-V to generate a consolidated editing instrution encapsulating several editing points.
>
> It is important to note that we did not directly incorporate existing datasets into our new combined dataset. Instead, we implemented a rigorous data curation process to ensure quality and consistency, which included the following steps:
>
> **1. Task Description and Ground-Truth Conversion:**
>
> - Generated a task-specific prompt to describe each dataset.
> - Converted ground-truth labels into natural language answers for consistency.
>
> **2. Quality Assessment of Candidate Datasets:**
>
> - Sampled approximately 100 entries from each candidate dataset.
> - Evaluated the quality and relevance of the dataset for training.
> - Rejected datasets deemed low quality or irrelevant.
>
> **3. Filtering of Selected Datasets:**
>
> - Removed samples that did not meet token length requirements (e.g., entries with fewer than two images or more than 15 images).
>
> **4. Dataset Splitting:**
>
> - Divided the newly collected dataset into training and validation subsets.
>
> **5. Integration and Performance Evaluation:**
>
> - Trained the new dataset alongside previously verified datasets.
> - Evaluated its impact on performance for both existing and new tasks.
> - Revisited data samples if the new dataset negatively affected performance or exhibited poor validation performance.
>
> **6. Conflict Resolution and Refinement:**
>
> - Resolved task/prompt conflicts between the new and existing datasets.
> - Double-checked data quality during this process.
>
> **7. Rebalancing Dataset Sizes:**
>
> - Adjusted the representation of datasets to ensure proportional and meaningful contributions to the combined dataset.
>
> Following these procedures, the final dataset was rigorously optimized to maintain performance on previous tasks while excelling on new tasks.
>
> Regarding **potential biases**, all our data were collected from publicly available datasets, and we did not impose any preferences that might exacerbate biases. However, we recognize the importance of addressing biases in datasets and will investigate this issue further in future work.
>
> We believe that this new dataset represents a valuable resource for future research, thanks to its enhanced quality, diversity, and comprehensive nature.

---

> ### Author Response · Authors · 2024-11-20
> **Response to the reviewer [Concern 2]**
>
> **Computational efficiency.**
>
> ---
>
> Our model **training** was conducted using 32 A100 GPUs.
>
> For **inference** speed, we note that since the vision encoder is relatively small compared to the language model, the primary factors affecting performance are the **size of the language model** and the **number of input tokens**. The model's complexity scales with the number of input tokens, which includes both image and language tokens. Thus, inference speed and model complexity are more closely tied to token count than to other factors.
>
> To evaluate the inference speed for multi-image scenarios, we based our analysis on the Qwen-7B model, using an average of six images as input. We compared this with two existing models: the **single-image model LLaVA-Next-Image** and the **multi-image model Mantis**. The inference speed results are provided below.
>
> It is worth noting that LLaVA-Next-Image employs the AnyRes technique, which typically divides an image into four patches and combines them with the original image, effectively processing five images in total. Despite this, our results indicate that our inference speed is comparable to existing models. Moving forward, we aim to explore more efficient methods for processing multiple images to further optimize performance.
>
> | Model | Language model | #Images | #Tokens Per Image | #Total tokens | Seconds/Sample |
> |-------------------|-------------------|----------------------------------|-----------------------------|---------| ---------|
> |LlaVa-Next-Image|Qwen-7b| 1|729x5| 3782 | 2.3
> |LlaVa-Next-Interleave|Qwen-7b| 6|729| 4470 | 2.5
> |Mantis|LLaMa-8b| 6|576| 3552 | 2.1|

---

> ### Author Response · Authors · 2024-11-20
> **Response to the reviewer [Concern 3-5]**
>
> **Scope of evaluation metrics.**
>
> ---
>
> - We conducted qualitative evaluations to assess the emerging capabilities of our model, as presented in **Tables 9–12 and 17** in our paper.
> - Additionally, we have included new examples in the appendix of our revised submission (**Tables 18–20**) to further illustrate these capabilities. While a user study would provide additional insights, due to time constraints, we plan to include such studies in future research.
>
> **Scalability.**
>
> ---
>
> We scaled our model from 0.5B to 14B parameters, with the 14B model representing a robust, large-scale LLM. To illustrate the scaling law, we created a figure (**Figure 6, appendix of the revised submission**) that demonstrates the model's performance across multi-image evaluations for both in-domain and out-domain metrics, averaged across tasks. The results indicate that scaling up LLMs significantly enhances performance. However, due to limited time and computational resources, further scaling will be explored in future research.
>
> **5. Generalization to Unseen Tasks.**
>
> ---
>
> We highlighted emerging capabilities of our model through qualitative visualizations.
> - For unseen tasks, we conducted **evaluations on out-domain data**, as shown in Table 1 of our paper. The results demonstrate that our model achieves state-of-the-art performance compared to prior models. However, **defining "completely unseen" tasks is inherently challenging**, and qualitative evaluations are limited by these ambiguities.
> - For now, we demonstrate the model's capabilities on certain unseen tasks via **visualization** (**Tables 9–12 and 17**) and have included additional examples in the appendix of our revised submission (**Tables 18–20**). We will continue to expand this type of evaluation in future research.
>
> Regarding **unseen modalities**, our current focus is solely on vision and language modalities. Research on other modalities is planned for future work.

---

### Official Review · Reviewer_5CHq · 2024-10-27

**Soundness:** 3
**Presentation:** 3
**Contribution:** 3
**Rating:** 8
**Confidence:** 2

**Summary:**

The paper proposed llava-next-interleave, which unifies the muliple image, multiple frame (video), mulitple view (3D), and Multiple patches (single-image) scenarios in one MLLM, which is very interesting.
One dataset namely M4-Instruct dataset and the LLava-interleav Bench is proposed, which is very helpful for pushing the progresses of the MLLM research.
Detailed ablation studies are provided, where some useful and meaningful insights are provided.

**Strengths:**

The paper makes attempt to unify the visual format, such as  multiple images, multiple frames, multiple views, and mulitple patches into one signal MLLM, which can facilitate the application of MLLM and improve generalization ability of MLLM on processing different format visual signals. With such unification, different visual formats only need to be processed with the corresponding formate, which can thereby be learned and tuned on the MLLM.

The composed dataset and benchmark are very useful for the research of MLLM.

Some ablation studies are conducted which are very interesting, such as the mixed interleaved data formats during training, as well as combining differenet data scenarios improves individual task performance.

**Weaknesses:**

Some insights may be further studied.

1. Mixed interleaved data formats can help improvign the perfromances. I am wondering the underlying reasons. One reason may be the different formats increases the diversity of the data. One reason may be the deficiency of the data, with mixed formats, the data are trained in two rounds (if I understand correctly), the performances are improved. The authors are highly suggested to performed detailed studies.

Please specify the experiment settings and also provide the results with the data with both formats. In this case, we can find with the data sufficiency matters for the performance gain.


2. The constructed data are trained in the continual training stage. I am wondering if we want to improve the ability of MLLM for handling different visual formats. Should we add more data (different types of visual formats) in the stage-i pretraining, besides the single image-text pair. Or we only perform the training in the continual training stage, the MLLM ablity on handling on multiple images, patches, frames, views can be well improved.

If possible, add the data in the pretraind stage. I am wondering whether there are still performance gain.

**Questions:**

Check detailed information in the Weakness part.

---

> ### Author Response · Authors · 2024-11-20
> **Response to the reviewer [Concern 1]**
>
> Thank you for your thoughtful comments and acknowledgment of our work. We have thoroughly responded to your feedback to address your concerns.
>
> **Mixed interleaved data formats.**
>
> ---
>
> We apologize for not providing sufficient details about our use of mixed data formats.
> - In our methodology, each data sample is **randomly assigned** to either an interleaved format or a front-loaded format, rather than being trained in both formats simultaneously. This approach ensures that the total number of training samples remains constant.
> - The observed improvement in performance is not a result of increased data quantity but rather the **enhanced diversity** introduced by this mixed-format strategy.
>
> In addition, we conducted **an ablation study to evaluate the impact of data sufficiency on multi-image performance**.
> - **Setting**: We randomly sampled **25%, 50%, and 75%** of the M4-Instruct dataset to fine-tune the LLaVA-NeXT-Image model.
> - **Results**: As summarized in the table below, it demonstrates that as the data size increases, the model exhibits strong scaling performance, highlighting the high quality and effectiveness of our M4-Instruct dataset. We also observe that, **the performance improvement from 0% to 50% is more substantial** than the improvement from 50% to 100%.
> - **Observation**: The results suggest that, for multi-image capabilities, **the enhancement of data diversity** (achieved in the 0% to 50% range) in M4-Instruct is more critical than merely increasing data quantity.
>
> M4-Instruct |Avg |Spot the Difference | Image Edit Instruction |Visual Story Telling |Text-rich VQA |Multi-image VQA |Puzzle|Q-Bench|NLVR2|
> |-------------------|-------------------|----------------------------------|-----------------------------|---------|---------|---------|-------------------------------|---------|---------|
> 0%|32.4 |12.9 |13.2|10.1| 59.6 |39.4 |9.0 |51.0| 68.0
> 25%|47.9|27.5|17.6|26.3|64.3|70.3|30.9|66.3|80.3
> 50%|52.8|32.8|20.3|28.3|69.2|77.8|41.2|69.2|84.2
> 75%|56.3|35.6|22.8|30.6|73.8|82.7|44.5|72.7|88.2
> 100%|58.6| 37.1| 24.3| 33.1| 76.1| 87.5| 48.7| 74.2| 88.8

---

> ### Author Response · Authors · 2024-11-20
> **Response to the reviewer [Concern 2]**
>
> **Continue training from image model or joint training.**
>
> ---
>
> Thank you for the valuable suggestion. We have conducted an ablation experiment to compare **direct training with all data formats against continued training from a single-image pre-trained model**, and we provide the detailed results below on single-image, multi-image, and video. In this table,
> - **The first row** represents direct training with a combination of single and multi-image data
> - **The second row** illustrates fine-tuning from single-image models (stage-i) using multi-image data and a subset of single-image data.
> - **The results clearly indicate that continual training yields superior performance**.
>
>
> | Continue Training | Multi-image   Benchmarks                  |                             |         |         |         | Single-image  Benchmarks                |         |         |         |         |                             |             Video  Benchmarks              |         |                             |             |       |         |          |             |
> |-------------------|----------------------------------|-----------------------------|---------|---------|---------|-------------------------------|---------|---------|---------|---------|-----------------------------|-----------------------------|---------|-----------------------------|-------------|-------|---------|----------|-------------|
> |                   | Mantis-Eval | BLINK | Q-Bench | NLVR2 | ScanQA | ai2d | chartqa | docvqa | MME* | pope | scienceqa_img | ActivityNet-QA (Acc/Score) | MVBench | Video Detailed Description | VideoChat-Correctness | VideoChat-Detail | VideoChat-Context | VideoChat-Temporal | VideoChat-Consistency |
> | from pretrain          | 41.0        | 37.6  | 47      | 54.0  | 27.7   | 46.3 | 38.3    | 47.5   | 47.1 | 85.4 | 59.4          | 44.7/2.17                 | 43.0    | 2.96                      | 2.97        | 2.87  | 3.49    | 2.42     | 3.14        |
> | from stage-i           | 45.6        | 39.2  | 52      | 67.8  | 29.3   | 52.2 | 52.2    | 59.2   | 52.0 | 86.8 | 60.6          | 48.0/2.84                 | 45.6    | 3.25                      | 3.12        | 2.97  | 3.62    | 2.36     | 3.27        |

---

> ### Comment · Reviewer_5CHq · 2024-11-26
> **Reply to the authors**
>
> The authors have done additional experiments and addressed my concerns. I am standing on my previous rating.

---

### Official Review · Reviewer_bKSe · 2024-10-30

**Soundness:** 3
**Presentation:** 3
**Contribution:** 2
**Rating:** 6
**Confidence:** 5

**Summary:**

The authors collect a interleave-formatted dataset from existing multi-image, video, multi-view, and image datasets. After that, they finetune a LMM from an existing single-image LMM to support multiple multi-image scenarios based on their collected dataset. Moreover, they curate a multi-scenario interleave benchmark to comprehensively evaluate LMM’s interleave capabilities.

**Strengths:**

In addition to the clear writing and logical flow of this work, I have outlined its strengths below:

1. This work presents a solid benchmark for comprehensively evaluating the interleaving capabilities of LMMs. This does help the LMM community.

2. The experiments in this work are thorough, providing an excellent baseline for future research in this field.

**Weaknesses:**

In my view, the technical innovation of this work is limited. I have outlined more detailed reasons below.

1. The proposed M4-Instruct dataset is merely a combination of several existing datasets, lacking sufficient technical innovation in its construction process.

2. The model proposed in this paper adopts the architecture and even checkpoints of previous methods, with only fine-tuning applied to the collected dataset. Technical innovation in the training process is also quite limited.

3. Although this work primarily follows a path of data collection and model fine-tuning, it only validates the data's effectiveness on a single model, LLaVA-Next. The authors should supplement their study by applying the M4-Instruct data to other 7B-sized single-image LMMs, such as InternVL-1.5, Cambrian, and MiniCPM-V2.5.

**Questions:**

See weakness above

---

> ### Author Response · Authors · 2024-11-20
> **Response to the reviewer [Concern 1]**
>
> We sincerely appreciate your valuable comments and recognition of our work. We have provided detailed responses to your comment and updated the relevant content in the revised manuscript, hoping to address your concerns.
>
> **Technical innovation in data.**
>
> ----
>
> We apologize for any lacking details about the technical innovation in our data. In the M4-Instruct dataset, we not only aggregated existing datasets but also created new ones through innovative methods, such as leveraging GPT-4 Vision (GPT-4-V) prompts. For instance, notable contributions is the construction of the MMChat_Twitter_Post, ScanQA, and MagicBrush dataset.
>
> - **For the MMChat_Twitter_Post dataset**, we utilized images sourced from an existing social media dataset MMChat. These images were input into GPT-4-V, paired with a language prompt specifically designed to generate Twitter-like posts. This process allowed us to create realistic and contextually relevant data entries, enhancing the overall dataset quality.
>
> - **For the ScanQA dataset**, since the original ScanQA only includes paired question-answer data with 3D point clouds, we sourced the previous ScanNet dataset and identified the mapping relationship between each point cloud and its corresponding video. From each video, we uniformly sampled 16 frames and prompt GPT-4-V to refine the aligned question-answer pairs in ScanQA, constructing multi-image instructions.
>
> - **For the MagicBrush dataset**, we transformed its original image editing task into various conversational formats. In general, we concatenate the source and target image in the instruction as the condition and set the task as predicting the corresponding editing prompt. For cases where multiple edits are required between two images, we either structure the data into multi-turn conversations, enabling models to progressively interpret the differences, or prompt GPT-4-V to generate a consolidated editing instrution encapsulating several editing points.
>
> It is important to note that **we did not directly incorporate existing datasets into our new combined dataset**. Instead, we implemented a rigorous data curation process to ensure quality and consistency, which included the following steps:
>
> **1. Task Description and Ground-Truth Conversion:**
>
> - Generated a task-specific prompt to describe each dataset.
> - Converted ground-truth labels into natural language answers for consistency.
>
> **2. Quality Assessment of Candidate Datasets:**
>
> - Sampled approximately 100 entries from each candidate dataset.
> - Evaluated the quality and relevance of the dataset for training.
> - Rejected datasets deemed low quality or irrelevant.
>
> **3. Filtering of Selected Datasets:**
>
> - Removed samples that did not meet token length requirements (e.g., entries with fewer than two images or more than 15 images).
>
> **4. Dataset Splitting:**
>
> - Divided the newly collected dataset into training and validation subsets.
>
> **5. Integration and Performance Evaluation:**
>
> - Trained the new dataset alongside previously verified datasets.
> - Evaluated its impact on performance for both existing and new tasks.
> - Revisited data samples if the new dataset negatively affected performance or exhibited poor validation performance.
>
> **6. Conflict Resolution and Refinement:**
>
> - Resolved task/prompt conflicts between the new and existing datasets.
> - Double-checked data quality during this process.
>
> **7. Rebalancing Dataset Sizes:**
>
> - Adjusted the representation of datasets to ensure proportional and meaningful contributions to the combined dataset.
>
> Following these procedures, the final dataset was rigorously optimized to maintain performance on previous tasks while excelling on new tasks. Therefore, we believe the data curation itself is also innovative.

---

> ### Author Response · Authors · 2024-11-20
> **Response to the reviewer [Concern 2]**
>
> **Technical innovation during training.**
>
> ---
> Thanks for this insightful question. We summarize the response as below.
>
> 1. Unified Training Formats:
>
> - Our main contribution is to integrat diverse data formats, including multi-image, video, 3D, and single-image, into a single interleaved training framework.
>
> 2. Preservation of Model Architecture for Ease of Adaptation:
>
> - We achieved the unification without modifying the existing model architecture, ensuring compatibility and adaptability across other models.
>
> 3. Simplicity as a Strength:
>
> - We prioritized simplicity over unnecessary complexity, providing practical advantages in implementation and scalability.
>
> 4. Recognition of Novelty:
>
> - The novelty and effectiveness of the unified framework were acknowledged by Reviewer jqfW (Strength 1).

---

> ### Author Response · Authors · 2024-11-20
> **Response to the reviewer [Concern 3]**
>
> **Applying the M4-Instruct data to other 7B-sized single-image LMMs.**
>
> ---
>
> Thank you for this valuable question. Given that many multi-modal LLMs adopt a LLaVA-like architecture, we first conducted a comparison between Qwen and LLaMA within our model framework. Additionally, as some models you mentioned do not have open-source training code, we performed an ablation study on ShareGPT4V and SPHINX by training it with our M4-Instruct data. The results, as shown below, demonstrate that our constructed dataset provides substantial benefits and utility when applied to other models.
>
> | Model | Lanuage Model | Setting | Spot the Difference|Image Edit Instruction |	Visual Story Telling	|Text-rich VQA|	Multi-image VQA| Puzzle|	Q-Bench |	NLVR2 |
> |-------------------|-------------------|----------------------------------|-----------------------------|---------|---------|---------|-------------------------------|---------|---------|---------|
> |LLaVA-Next-Image|Qwen-7b|Zero-shot|12.9 |13.2 |10.1| 59.6 |39.4 |9.0 |51.0 |68.0|
> |LLaVA-Next-Interleave|Qwen-7b|SFT with M4-Instruct|37.1 |24.3 |33.1 |76.1 |87.5 |48.7 |74.2| 88.8|
> |LLaVA-Next-Image|LLaMA-8b|Zero-shot|17.1|11.4|10.2|60.1|45.5|9.6|51.7|68.8
> |LLaVA-Next-Interleave|LLaMA-8b|SFT with M4-Instruct|36.8|24.6|33.4|76.0|86.9|48.5|73.6|86.3
> |SPHINX-MoE-Image|Mixtral-8×7B|Zero-shot|8.4|9.7|8.8|42.3|31.9|9.5|40.2|54.2|
> |SPHINX-MoE-Interleave|Mixtral-8×7B|SFT with M4-Instruct|26.5|18.7|26.2|70.3|75.4|44.9|64.1|82.7
> |ShareGPT4V-Image|Vicuna-7B|Zero-shot|10.3|11.2|10.1|45.5|34.3|9.2|42.8|52.3|
> |ShareGPT4V-Interleave|Vicuna-7B|SFT with M4-Instruct|32.4|25.4|27.8|66.8|79.8|48.8|68.4|80.6

---

> > ### Comment · Reviewer_bKSe · 2024-11-22
> > **To the authors**
> >
> > Thanks for your responses. My main concerns are addressed. So I consider to raise my score to 6.

---

### Meta-Review · Area_Chair_GZY2 · 2024-12-18

**Metareview:**

This paper proposed LLaVA-NeXT-Interleave which is a unified model to tackle multi-image, multi-frame, multi-view and multi-patch scenarios in LLMs. This paper curated a M4-Instruct dataset containing around 1M samples across 4 primary domains with14 tasks. The developed unified model is comprehensively evaluated and showed improvement compared with prior art on multi-image, multi-frame and multi-view tasks.

Strength:
1. As agreed by all reviewers, the proposed M4-Instruct dataset and benchmark are solid and comprehensive and are useful for MLLM research.
2. The unification of the framework is innovative.
3. The experiments are thorough.
4. Achieving state-of-the-art performance.

Weakness:
1. Reviewers proposed some weakness the authors addressed them in the discussion. Reviewer bKSe and 5CHq mentioned their concerns are addressed and concerns from reviewer jqfW are also answered.
2. The template used for this paper is slightly different from the template of ICLR draft. As the difference is small, we do not take this into consideration while making decisions. However, the authors should pay attention to the template usage for paper submission.

**Additional Comments On Reviewer Discussion:**

Reviewer bKSe mentioned the following weakness of the paper:
1. There is not enough technical innovation in constructing the M4-Instruct dataset.
2.  No technical contribution to the model development.
3.  Method only tested on a single model.
The authors provided more details of their effort of curating the datasets to address point 1. They also emphasis the novelty of model training including unified training formats and preservation of model architecture to address point 2. The authors applied proposed dataset to other models to address point 3. The reviewer acknowledged the concerns have been addressed.

Reviewer 5CHq pointed out that:
1. This paper does not have the insights about why proposed dataset could help improve the performance.
2. Whether use the proposed dataset during pre-training would also improve the performance.
The authors addressed point 1 by providing additional experiments demonstrating the improvement is from data diversity. They also provide an additional experiment to show that using proposed dataset in continue training is better than using them in pre-training stage. The reviewer acknowledge the concerns have been addressed.

Reviewer jqfW asked the authors to provide more details, including:
1. Details in data curation.
2. Discussion about computation efficiency.
3. Scope of evaluation.
4. Scalability and practical deployment.
The authors provided more details about the above four points. From my point of review, these details could address these concerns.

---

### Decision · Program_Chairs · 2025-01-22

Accept (Spotlight)